# HDAC5 Inhibitors as a Potential Treatment in Breast Cancer Affecting Very Young Women

**DOI:** 10.3390/cancers12020412

**Published:** 2020-02-10

**Authors:** Sara S. Oltra, Juan Miguel Cejalvo, Eduardo Tormo, Marta Albanell, Ana Ferrer, Marta Nacher, Begoña Bermejo, Cristina Hernando, Isabel Chirivella, Elisa Alonso, Octavio Burgués, Maria Peña-Chilet, Pilar Eroles, Ana Lluch, Gloria Ribas, María Teresa Martinez

**Affiliations:** 1INCLIVA Biomedical Research Institute, Hospital Clínico Universitario Valencia, University of Valencia, 46010 Valencia, Spain; sara.oltra@uv.es (S.S.O.); juanmitch5@hotmail.com (J.M.C.); eduardo.tormo@uv.es (E.T.); marta.albanell.95@gmail.com (M.A.); ferrermartineza9@gmail.com (A.F.); martanacher13@gmail.com (M.N.); begobermejo@gmail.com (B.B.); c.hernandomelia@gmail.com (C.H.); chirivella_isa@gva.es (I.C.); mariapch84@gmail.com (M.P.-C.); pilar.eroles@uv.es (P.E.); lluch_ana@gva.es (A.L.); 2Biomedical Research Centre Network in Cancer (CIBERONC), 46010 Valencia, Spain; elisa.alonso@uv.es (E.A.); octavioburgues@gmail.com (O.B.); 3Pathology Department, Hospital Clínico Universitario Valencia, University of Valencia, 46010 Valencia, Spain

**Keywords:** breast cancer, young women, histone deacetylase, HDAC5 inhibitors, LMK-235

## Abstract

Background: Breast cancer in very young women (BCVY) defined as <35 years old, presents with different molecular biology than in older patients. High *HDAC5* expression has been associated with poor prognosis in breast cancer (BC) tissue. We aimed to analyze *HDAC5* expression in BCVY and older patients and their correlation with clinical features, also studying the potential of HDAC5 inhibition in BC cell lines. Methods: *HDAC5* expression in 60 BCVY and 47 older cases were analyzed by qRT-PCR and correlated with clinical data. The effect of the HDAC5 inhibitor, LMK-235, was analyzed in BC cell lines from older and young patients. We performed time and dose dependence viability, migration, proliferation, and apoptosis assays. Results: Our results correlate higher *HDAC5* expression with worse prognosis in BCVY. However, we observed no differences between *HDAC5* expression and pathological features. Our results showed greatly reduced progression in BCVY cell lines and also in all triple negative subtypes when cell lines were treated with LMK-235. Conclusions: In BCVY, we found higher expression of *HDAC5*. Overexpression of *HDAC5* in BCVY correlates with lower survival rates. LMK-235 could be a potential treatment in BCVY.

## 1. Introduction

Breast cancer is the most common cancer among women worldwide [1]. As a percentage of all cancers in young women, breast cancer rates increase during the third and fourth decades of life, from 2% at age 20 to more than 40% by age 40. This abrupt increase is attributable to routine screening mammography. Breast cancer incidence is similar among young women in both developed and developing countries [2], but in recent years, there has been an increase in breast cancer diagnoses in very young women [3]. The distribution of breast cancer subtypes and grades changes with age, with more aggressive phenotypes among young women compared with older counterparts [4]. Basal-like and Her2-enriched breast tumors are also more common in young than in older women [5]. Scientific evidence suggests that age at breast cancer diagnosis represents an independent prognostic survival factor [6]. Indeed, multiple studies conclude that early age at diagnosis of breast cancer is highly associated with an increased risk of recurrence and death [7].

Epigenetic modifications are reported to play an important role in the onset and progression of many diseases. Carcinogenesis can be explained by genetic alterations, but epigenetic processes such as DNA methylation, histone modifications, and non-coding RNA deregulation are also involved [8]. 

Histone deacetylases are essential for global acetylation patterns in the nucleus and the epigenetic regulation of gene expression [9]. Histone deacetylases (HDACs) have been stratified into classes I, IIA, IIB, III, and IV and increased expression of these isoenzymes has been observed in different tumors, also often associated with poor outcome [10,11,12,13]. It has been reported several times that their levels vary greatly in cancer cells and differ according to tumor type. Of the 18 human HDACs, HDAC5 (a class IIa HDAC) is involved in synoviocyte activation, neural regeneration and repair, differentiation of myoblasts, and recently, elevated expression of *HDAC5* has been correlated with worse prognosis in patients with breast cancer (BC) [14].

Histone deacetylase inhibitors (HDACi) are currently acquiring importance in cancer treatment, being the only approved epigenetic therapy in clinics altering histone proteins to date. HDACi have been classified into four groups depending on their structure: Hydroxamic acids, cyclic peptides, benzamides, and short chain fatty acids. The hydroxamic acids (vorinostat, belinostat, and panobinostat) have been approved by the FDA as anticancer treatments. However, they are limited by their nonspecificity, affecting all HDACs [15]. These are known as pan-HDAC inhibitors and may also cause numerous side effects due to their broader specificity. Therefore, in some cases, more selective inhibitors may be more effective in therapy [16,17,18,19].

A previous study based on evaluation of new HDAC inhibitors identified LMK235 (N-((6-(hydroxyamino)-6-oxohexyl)oxy)-3,5-dimethylbenzamide) as the most cytotoxic compound, displaying equipotent HDAC inhibition in pan-HDAC assay when compared to vorinostat. In contrast to vorinostat, LMK235 showed a novel HDAC isoform selectivity profile with a preference for HDAC4 and HDAC5 [20].

Previous group studies showed significant differences in miRNA expression [21] and methylation [22] profiles among breast cancer affecting very young women (<35 years) (BCVY) and breast cancer in older patients (>45 years) (BCO). Additionally, our group observed considerable hypomethylation of CpG regions that were regulating *HDAC5* expression in BCVY, and this methylation was related to significant *HDAC5* overexpression in BCVY patients [23]. In the present study, we analyse *HDAC5* expression in a large cohort of BC patients, to clarify the correlation between *HDAC5* overexpression and relapse and survival in BCVY and BCO and the inhibitory effect of LMK-235 in breast cancer cell lines from very young and older women with BC. 

## 2. Results

### 2.1. Clinical-Pathologic Characteristics of Patients

A total of 107 patients were included, 60 were young women under 35 years (BCVY) and 47 samples from women over 45 years (BCO). The median age at breast cancer diagnosis in the young patient’s group was 32 years (range, 20–35), and in the old patient’s group was 69 years (range, 53–94). Hereditary cases with BRCA1 and BRCA2 mutations were excluded from the study. In BCVY cohort, the immunohistochemical analyses showed 39.9% (n = 24) of luminal patients, 11.6% (n = 7) HER2-positive, 21.6% (n = 13) luminal/HER2 and 23.3% (n = 14) triple negative. In BCO group, 59.5% (n = 28) of patients were luminal, 10.6% (n = 5) HER2-positive, 10.6% (n = 5) luminal/HER2 and 17% (n = 8) presented triple negative subtype. Median follow-up was 93.4 months (Table 1).

### 2.2. HDAC5 Overexpression in BCVY Patients Correlates with Poorer Prognosis

*HDAC5* was significantly overexpressed in BCVY patients (*p*-value = 0.04) (Figure 1A). We found no appreciable differences among clinical features and *HDAC5* expression, apart from tumor grade in BCO patients (*p*-value = 2.8 × 10^−3^) where the lower grades presented higher *HDAC5* expression (Appendix A). Regarding molecular subtypes, we observed higher *HDAC5* expression in Luminal B and HER2 tumors from BCVY in comparison with BCO patients from the same subgroups. While BCO patients presented higher expression for Luminal A and Luminal/HER2 comparing to BCVY (Appendix A). Despite no significant association, these results agree with higher expression of *HDAC5* for poor prognostic subtypes for each patient age groups in our cohort, that were HER2 for BCVY and Luminal/HER2 for BCO patients.

We observed similar percentages of cancer death between young and old patients, 13.8% (n = 9) and 13.3% (n = 6), respectively. However, *HDAC5* was significantly overexpressed in BCVY patients that died in comparison with BCO (*p*-value = 0.04) and a significant *HDAC5* overexpression was observed for BCO patients that survive when compare with BCO patients that died (*p*-value = 0.01) (Figure 1B). These results were in line with survival studies, where Kaplan–Meier curves showed a reduced survival trend for BCVY when *HDAC5* was overexpressed (Figure 2A) whereas BCO women presented poorer survival when *HDAC5* was repressed (Figure 2B). In terms of relapse, results were not as evident as survival (results not shown). For survival and relapse studies, samples were classified in high, medium or low, according to *HDAC5* expression. Despite no significant results observed for BCVY patients, survival studies indicated an important trend of worst survival for BCVY, contrary to the results observed for BCO. In this regard, *HDAC5* overexpression in BCVY could be related with the poorer outcome at this age group. It is worth to mention that the number of BC patients that experience relapse and/or death was reduced and further studies should be addressed in order to validate this tendency observed. 

### 2.3. LMK-235 Inhibitor Reduces Proliferation in BC Cell Lines

Breast cancer cell lines were exposed to increasing concentrations of HDAC5 inhibitor LMK-235 (0, 0.625, 1.25, 2.5, 5, 10, and 20 µM) for 24, 48, and 72 h. We observed that relative proliferation decreased in a dose- and time-dependent manner for some breast cancer cell lines (Figure 3). After 48 h treatment with low doses of LMK-235 (Figure 3A), cell viability of MDA-MB-231 and HCC1806 cell lines was severely compromised. Additionally, viability was notably diminished in the HCC1937 cell line after 72 h of treatment, showing similar results to MDA-MB-231 and HCC1806 cell lines (Figure 3B). Although no significant results were obtained for HCC1500 cell line after 48 h of treatment, the young cell line showed a 50% reduction in viability after 72 h of low dose LMK-235. Breast cancer cell lines that experienced significant viability reduction were triple negative subtype with the exception of HCC1500 luminal cell line established from young BC patients and showed higher reduction in comparison with the older luminal cell lines, MCF7 and BT474. Inhibitor studies reveal important response for triple negative BC cell lines to LMK-235 and for luminal cell lines from young women with BC not observed in older luminal cell lines. 

### 2.4. Reduced Migration of Breast Cancer Cell Lines Treated with LMK-235 Measured by Wound-Healing Assay

Wound-healing assays demonstrated that LMK-235 significantly inhibited migration in HCC1937 (*p*-value = 0.01), HCC1806 (*p*-value = 4.9 × 10^−7^) and MDA-MB-231 (*p*-value = 1.2 × 10^−3^) breast cancer cell lines after 48 h of treatment. Specifically, we observed ~23% less cell migration in the HCC1937 cell line, ~14.7% for MDA-MB-231, and also HCC1806 cell line showed a substantial reduction. HCC1500 cells treated with LMK-235 presented cell migration reduced by ~5.3%. However, the last cell line exhibited considerable cell death at 48 h of cell treatment making wound-healing analysis difficult. In contrast, breast cancer cell lines MCF-7 and BT474, both from luminal old BC patients, showed insignificant cell migration reduction when cells were treated with LMK-235 (Figure 4). 

These results revealed higher cell migration and proliferation reduction in cell lines from triple negative subtypes (MDA-MB-231, HCC1806, and HCC1937), independently of age. However, we observed markedly diminished viability in the young and luminal HCC1500 cell line treated with HDAC5 inhibitor, which could not be detected in the luminal subtypes from older cell lines (MCF7 and BT474). 

### 2.5. Apoptosis AfterLMK-235 Treatment

To confirm the results obtained by MTT, we performed an apoptosis assay by flow cytometry. After 48 h of treatment, similar results to the MTT assay were observed. Results show increasingly early apoptosis in most BC cells lines after 48 h of LMK-235 (10 and/or 20 µM) treatment. Interestingly, and in line with previous proliferation and migration results, the most important increase in apoptosis was observed in triple negative cell lines from both BCVY and BCO. Intriguingly, LMK-235 induced a more profound increase in HCC1500 cell death for higher concentrations than in MTT assay (Figure 5A). These results agree with the wound-healing assay observations, where after 48 h of treatment most of the HCC1500 cells were dead and no wound-healing analysis could be performed. 

### 2.6. Accumulation of Acetyl-H3 After LMK-235 Treatment in Breast Cancer Cell Lines

We observed increased *HDAC5* mRNA expression in all breast cancer cell lines that were treated with LMK-235 (Figure 5B). These results confirm that the HDACi does not act at the mRNA level, but rather at the protein level. LMK-235 induces an increase in mRNA expression as a positive feedback mechanism in order to increase and restore protein expression. Additionally, *HDAC5* expression was higher in control/DMSO conditions for all BC cell lines independently of their response to the HDAC5 inhibitor. 

To investigate whether the changes in *HDAC5* mRNA expression are reflected in persistent changes at their protein levels, these were determined in breast cancer cell lines by western-blotting following treatment with LMK-235 (20 µM) for 48 h. The levels of HDAC5 protein were barely detectable for all cell lines, as previously observed in other studies [14,24] that showed the low quantity of HDAC5 protein by western blot (data not shown). Next, we measured HDAC5 activity at the acetylation of lysine residues on histone H3 to determine the inhibitory effect of LMK-235 on HDAC5. Western-blot studies showed accumulation of acetyl-histone H3 in breast cancer cell lines after 48 h of LMK-235 treatment (Figure 5C,D). We observed acetyl-histone H3 accumulation both in luminal breast cancer cell lines (BT474, HCC1500 and MCF7) and in triple negative breast cancer cell lines (HCC1937 and MDA-MB-231). These results demonstrate that LMK-235 specifically inhibits HDACs, which catalyze the removal of acetyl groups from N-acetyl lysine histone residues. Therefore, their inhibition produces an accumulation of acetyl-histones and acetyl-H3 is one of the most extensively modified.

## 3. Discussion

BCVY can be understood as a unique and different entity from breast cancer in older people, not only because of patients’ characteristics (where aspects such as fertility preservation must take into account) but also in its biological and molecular tumor characteristics. Our group has focused for years on the study of these differential molecular characteristics in breast cancer tumors of young and old women. Previous results [23] showed us that *HDAC5* was notably overexpressed in BCVY. We validate this issue analyzing *HDAC5* overexpression in a larger cohort of young patients under 35 years and over 45, as well as studying the correlation of *HDAC5* expression levels with different clinic-pathological features. Additionally, we analyzed the effects of an HDAC5 inhibitor, LMK-235, on apoptosis, proliferation, and migration in breast cancer cell lines.

Histones are acetylated on lysine side chains, which neutralizes lysine’s positive charge, leading to open chromatin structure by reducing interaction between histone and negatively charged DNA [25,26]. Thus, histone acetylation increases the accessibility of proteins such as transcription factors to promoters and enhancers, thereby mediating active gene expression. Acetylated histones also function as binding sites for numerous proteins with bromodomains, which often activate gene transcription [25]. In contrast, histone deacetylation is associated with chromatin condensation and transcriptional repression [25]. Analogous to histone methylation, histone acetylation is reversibly controlled by two large enzyme families: histone lysine acetyltransferases (KATs) and HDACs.

Turning to *HDAC5*, high expression has been correlated with poorer prognosis in patients with BC or pancreatic neuroendocrine tumors and has been attributed to oncogenic effects [27,28]. Specifically, *HDAC5* overexpression has been correlated with triple negative BC tumors [14,29]. However, our study showed higher *HDAC5* expression in BCVY than BCO samples. Next, we wonder whether *HDAC5* overexpression was correlated with relapse and survival, and other clinical and pathological data. BCVY patients with *HDAC5* overexpression showed a higher risk of exitus, whereas results were contrary for BCO that showed a significant risk of death when *HDAC5* was repressed. In agreement with these results, higher *HDAC5* expression was observed in BCVY samples that died whereas lower expression was found in BCO patients that die. Our results reinforce the hypothesis that *HDAC5* overexpression in BCVY patients is related with a poor outcome observed in this group of age. BCVY is not a usual diagnosis, so there is a limitation in the number of BCVY samples and the present study is actually the first report that includes one of the highest cohort of BCVY patients in comparison with other larger datasets (ex., TCGA and METABRIC [30,31]). 

Next, we evaluated *HDAC5* expression and its correlation with different clinical and pathological features (histologic subtypes, Ki67, tumor size, grade, and axillary affection). We found no correlation between *HDAC*5 expression and clinical features in BCVY. The only meaningful association observed was between *HDAC5* downregulation and higher tumor grades (II and III) in BCO. Higher tumor grades are associated with poorer prognosis and in our study with *HDAC5* repression for BCO, which agrees with survival tendencies observed at this age group where *HDAC5* repression correlates with poorer survival in BCO. 

HDAC inhibitors are the most widely investigated epigenetic drugs in clinical studies [32]. In vitro studies suggest that the HDAC5 inhibitor LMK-235 could be a novel therapeutic strategy for BC treatment [14]. We evaluated the inhibitory effect of LMK-235 in breast cancer cell lines from young women and compared it with breast cancer cell lines from older counterparts. Our results demonstrated that LMK-235 induced important reduction in progression, migration, and apoptosis, not only in triple negative young and old but also in luminal young cell lines. Regarding that, we observed important viability reduction under treatment conditions for the cell lines that presented significant differences in the migration assay between control and treatment conditions. These results were observed in triple negative cell lines (from young and old patients). Thus, the reduction in cell viability correlates with a reduction in cell migration under LMK-235 treatment. Interestingly, the luminal cell lines from old patients (BT474 and MCF7) were not affected by LMK-235 inhibitor. However, the luminal young cell line HCC1500 presented intermedium results between triple negative and luminal old cell lines. These results point out a potential breast cancer treatment not only for triple negative breast cancer but also for young patients from different molecular subtypes.

Additionally, LMK-235 response was lower in ER-positive cell lines, except for HCC1500 which presented an intermedium response in comparison with the rest of ER-positive cells, as we previously mentioned. Previous studies found LMK-235 response in triple negative BC cell lines [33,34,35,36]. However, our results showed an important response to LMK-235 in the breast cancer cell line from young women that present luminal subtype, as HCC1500. These results suggest an effect of ER in the LMK-235 treatment but must exist other mechanisms in young cell lines that increase the effect of LMK-235, which are not present in older cell lines. There was a pronounced time- and dose-dependent decrease in cell viability and migration and hence an increased apoptosis in all breast cancer cell lines from triple negative subtypes (BCVY and BCO). Furthermore, HCC1500 luminal BCVY cell line presented similar results, showing significant apoptosis rates and reduced cell viability, avoiding the possibility of analyzed migration wound-healing assays. It is worth to mention that LMK-235 treatment induces higher apoptosis in HCC1500 cell line at lower doses in comparison with the rest of cell lines. These results suggest the existence of off-target effects that increase the apoptosis at this cell line, so further studies are required to analyze the underlying mechanisms of LMK-235 inhibitor. 

These results reinforce the hypothesis that *HDAC5* is involved in tumorigenesis and cancer progression reducing survival, specifically in BCVY. We propose studying HDAC5 inhibitors like LMK-235 in a larger cohort of breast cancer cell lines from young and old women and from different molecular subtypes in order to evaluate their inhibitory effect. Additionally, cell lines studies show a significant response to HDAC5i of triple negative BCO in addition to all BCVY cell lines independently of their molecular subtype.

## 4. Materials and Methods 

### 4.1. Patient Selection

All samples included in the study were archived formalin-fixed paraffin-embedded (FFPE) BC tissues stored at the Hospital Clínico Universitario of Valencia, Spain. We selected 60 breast cancer samples from women under 35 years old (BCVY) and 47 samples from women over 45 (BCO). The clinical characteristics of patients are included in Table 1. This study was approved by the Institutional Health INCLIVA-Hospital Clínico Ethics Committee and informed consent was obtained from all subjects. The code of the ethical committee of the Hospital Clínico Universitario of Valencia (Spain) is 2013/128.

All patients’ clinical data and tumor pathological characteristics were collected at their first visit and information about relapse and survival on subsequent visits. Tumor size and lymph node involvement were assessed using the eighth edition of the American Joint Committee on Cancer staging manual [37]. Patient clinicopathological data used in this study are shown in Table 1.

### 4.2. Breast Cancer Cell Lines Culture and Treatment

Breast cancer cell lines were obtained from the American Type Culture Collection (ATCC, Rockville, MD, USA). Cell lines were cultured in RPMI 1640 or DMEM medium supplemented with 1% L-glutamine and 10% fetal bovine serum (GIBCO, New York, NY, USA). The culture conditions were identical in all cell lines: 37 °C and 5% CO_2_ (Table 2). Cells were seeded for 24 h before treatment with LMK-235 (Selleck Chemicals, Houston, TX, USA) or DMSO (control) for MTT assay, wound-healing, and apoptosis assay. 

### 4.3. Cell Proliferation Assay

The protocol used is based on a colorimetric MTT (3-(4,5-dimethylthiazol-2-yl)-2,5-diphenyltetrazolium bromide) assay. We seeded and cultured 2000 cells in 96-well plates. Next, cells were treated with a specific drug dose, and MTT assay was performed 24, 48, and 72 h after treatment. The absorbance of this colored solution was quantified by measuring at 590 by spectrophotometer. Each experiment was performed in triplicate and repeated at least twice. Average values for triplicates were calculated. Absorbance observed at different drug concentrations was compared with the respective non-treated controls and cell viability was calculated.

### 4.4. Scratch Wound-Healing Assay

Cells were seeded in six-well plates at a density of 5 × 10^5^ cells/well and incubated overnight until they reached 70% confluence. A pipette tip was used to generate a wound in the cell layer. Cells were then treated with LMK-235 (20 μM). Each experiment was performed in triplicate and repeated at least twice. Images were obtained at 0, 24 and 48 h at the same position and percentage of cell migration was evaluated using ImageJ (LOCI, University of Wisconsin).

### 4.5. Apoptosis Assay by Flow Cytometry

We plated 100,000 cells in 12-well plates. After 48 h of treatment, trypsinized and floating cells collected from the supernatant were centrifuged and incubated with Annexin V-FITC (BioLegend, San Diego, California, United States) and DAPI for 15 min. Apoptosis was detected using Flow Cytometry BD LSRFortessa™ (BD, Franklin Lake, NJ, USA).

### 4.6. Western Blot

We then prepared the nuclear protein fraction of breast cancer cell lines using a nuclear extraction kit (Active Motif, Belgium, Germany). A total of 40 µg of protein were resolved by 12% SDS-PAGE and transferred onto nitrocellulose membranes. Membranes were blocked in 5% BSA and hybridized with antibodies against aceyl-H3 (1:1000) and GAPDH (1:1000) as a loading control. Immunoreactive bands were visualized by chemiluminescence (GE Healthcare, Life Science, Oslo, Norway) and captured by Image Quant™ LAS4000 (GE Healthcare, Life Science, Oslo, Norway). The band densities, normalized to the GAPDH, were analyzed with ImageJ software.

### 4.7. RNA Extraction

Total RNA from BC cell lines was isolated using the High Pure RNA Isolation Kit (Roche, Basel, Switzerland), following the manufacturer’s protocol. Total RNA from FFPE samples was isolated using RecoverAll Total Nucleic Acid Isolation Kit (Applied Biosystems™ by Life Technologies™, Carlsbad, CA, USA), following the manufacturer’s protocol. RNA concentration was measured using a NanoDrop ND 2000-UV-vis Spectrophotometer (Thermo Fisher Scientific Inc., Wilmington, DE, USA).

### 4.8. Gene Expression by qRT-PCR

Gene expression of *HDAC5* and *GAPDH* as endogenous control was carried out by quantitative real time-PCR (qRT-PCR) in RNA extracted from FFPE samples. We used the TaqMan® Gene Expression Assays (Applied Biosystems™ by Life Technologies™, Carlsbad, CA, USA). Normalization was done with *GAPDH*. The data were managed using the QuantStudio Desing & Analysis Software v1.4 (Applied Biosystems™ by Life Technologies™, Carlsbad, CA, USA). Relative expression was calculated using the comparative Ct method and obtaining the fold-change value (ΔΔCt). 

### 4.9. Statistical Analysis

All statistical analysis was performed using R Bioconductor (https://www.bioconductor.org/). To determine differences between *HDAC5* expression and breast cancer patients/cells from different ages we performed a Wilcoxon Rank Sum test. Results were considered significant when *p*-value < 0.05. OS and RFS were performed using follow-up data from BC patients to analyze the prognostic value of *HDAC5* expression in terms of relapse and survival. Patients were divided into three groups according to the distribution of the log10 *HDAC5* Fold change: high expression (> 0.2 log10 Fold change), medium (> −0.2 and <0.2) and low (< −0.2). RFS and OS studies were performed using *survival* R package (https://CRAN.R-project.org/package=survival). 

## 5. Conclusions

Taken together, the results provide insights into the biology of breast cancer in younger women and support our hypothesis that breast tumors of younger women activate molecular pathways related to increased aggressiveness, such as *HDAC5*. Our findings are consistent with those previously published by Li et al. [14], confirming that *HDAC5* promotes proliferation, invasion, and migration in human breast cancer. We also demonstrate that *HDAC5* overexpression in BCVY correlates with lower survival rates.

In our study, we show that targeted treatment with HDAC5 inhibitor LMK-235 reduces migration and proliferation of tumor cells and increases apoptosis in triple negative breast cancer cell lines, as previous results demonstrated, and as a novelty in luminal cell lines from younger BC patients. In summary, our findings show, for the first time, a potential treatment specific for breast cancer affecting very young patients. However, more efforts should be addressed to analyze its therapeutic role. 

## Figures and Tables

**Figure 1 cancers-12-00412-f001:**
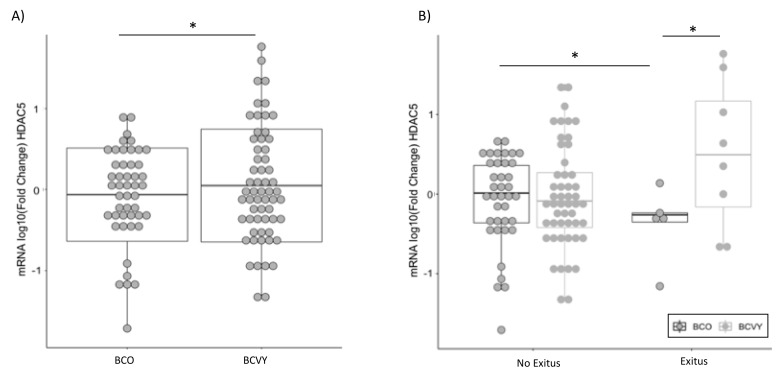
*HDAC5* expression in BCVY (n = 60) and BCO (n = 47) patients analyzed by qRT-PCR (**A**). *HDAC5* expression in BCVY and BCO patients regarding their status (exitus or no exitus) (**B**). BCVY: Breast cancer in very young women; BCO: Breast cancer in old women. * *p* ≤ 0.05, ** *p* ≤ 0.01.

**Figure 2 cancers-12-00412-f002:**
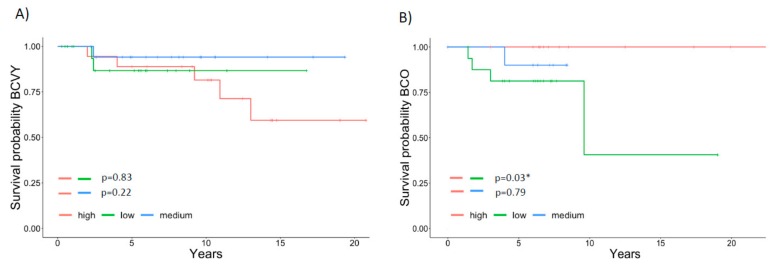
Association of *HDAC5* expression and survival in breast cancer patients. Survival curves for *HDAC5* expression in BCVY (n = 42) (**A**) and BCO patients (n = 28) (**B**). *HDAC5* expression was classified in high (red), medium (blue) or low (green). BCVY: Breast cancer in very young women; BCO: Breast cancer in old women. * *p* ≤ 0.05.

**Figure 3 cancers-12-00412-f003:**
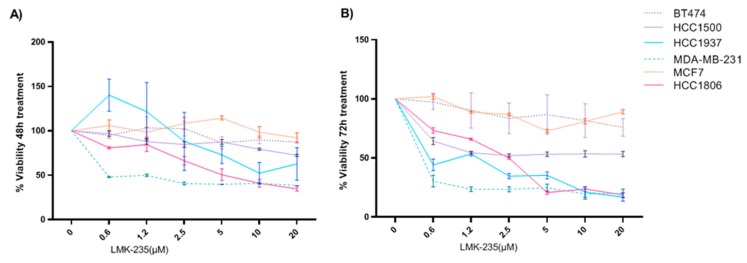
Effects of LMK-235 treatments in BC cell proliferation. Cell proliferation was determined by MTT assay for HCC1500 and HCC1937 (<35 years old) and MDA-MB-231, MCF-7, HCC1806, and BT-474 (>45 years old) cell lines that were treated with LMK-235 (0 to 20 μM) for 48 hours (**A**), 72 h (**B**). Points indicate the mean of at least three independent experiments and the standard deviation.

**Figure 4 cancers-12-00412-f004:**
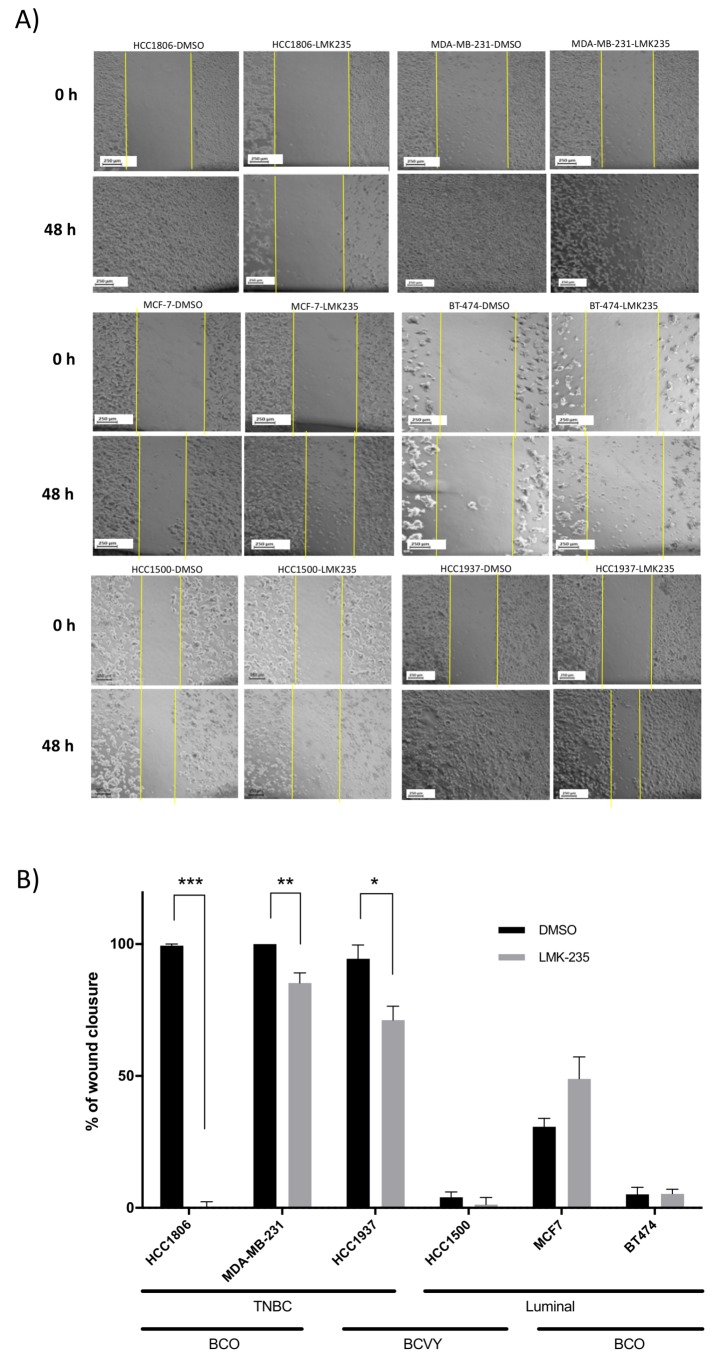
Effects of LMK-235 treatments in BC cell migration. Cell migration was assessed by scratch wound-healing assay after treatment with LMK-235 (20 μM) for 48 h. Images of cell migration at 0 h and 48 h after LMK-235 (20 μM) treatment (**A**); Percentage of wound closure after 48 h of LMK-235 (20 μM) treatment or DMSO/control. Three separate experiments were conducted, and representative results are shown (**B**). Columns mean ± SD of the percentage of wound closure in the three independent experiments. Black bars represent control DMSO cells and grey bars represent cells treated with LMK-235. * *p* ≤ 0.05, ** *p* ≤ 0.01, *** *p* ≤ 0.001 statistically significant. TNBC: Triple negative breast cancer; BCVY: breast cancer in very young women; BCO: Breast cancer in old women.

**Figure 5 cancers-12-00412-f005:**
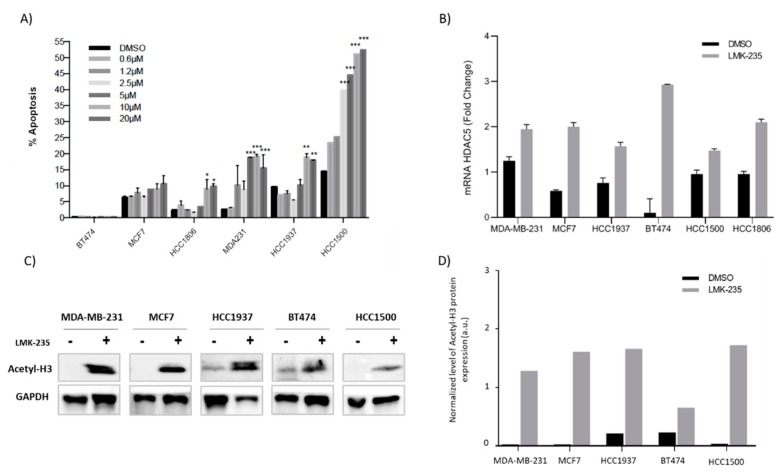
Percentages of apoptosis are shown after treatment with LMK-235 (0.6 to 20 μM) or DMSO for 48 h. Apoptosis was analyzed by triplicated and SD was calculated (**A**); Expression of *HDA5* in breast cancer cell lines treated with LMK-235 (20 μM) and DMSO control (**B**); Accumulation of acetyl-histone H3 after 48 h of LMK-235 (20 μM) treatment in breast cancer cell lines examined by western-blot. The levels of acetyl-histone H3 were determined by western blot. GAPDH was used as a loading control (**C**). Densitometry western bolt analyzed by ImageJ software (**D**). * *p* ≤ 0.05, ** *p* ≤ 0.01, *** *p* ≤ 0.001.

**Table 1 cancers-12-00412-t001:** Clinicopathological information of breast cancer (BC) samples and statistical results from the study of HDAC5 expression vs. clinicopathological features by age groups.

	BCVY	*p*-Value (HDAC5 ~ CP)	BCO	*p*-Value (HDAC5 ~ CP)
N	60		47	
Age mean (years ± SD)	32.1 ± 3.3		69.8 ± 9.3	
Histological subtypes (%)		0.46		0.40
Luminal A	16.6		27.6	
Luminal B	23.3		31.9	
TNBC	23.3		17.0	
Luminal/HER2	21.6		10.6	
HER2	11.6		10.6	
Unknown	3.6		2.3	
ER status (%)				
ER+	60.0	0.43	74.4	0.92
ER-	33.3		21.3	
Unknown	6.7		4.3	
PR status (%)				
PR+	50.0	0.44	57.4	0.44
PR-	43.3		38.3	
Unknown	6.7		4.3	
HER2 (%)				
HER2+	31.6	0.97	21.3	0.93
HER2-	61.6		74.5	
Unknown	6.8		4.2	
KI67 (%)		0.56		0.07
<15%	16.6		27.7	
15–30%	28.3		32.0	
>30%	38.3		27.7	
Unknown	16.8		12.6	
Grade (%)		0.16		2.8 × 10^−3^ **
I	13.2		23.3	
II	49.1		41.9	**
III	37.7		34.9	*
Unknown				
Tumour Size (%)		0.16		0.02 *
<2 cm	35.0		59.6	
2–5 cm	40.0		23.4	
>5 cm	20.0		8.5	
Unknown	5.0		8.5	
Axillary Affection (%)		0.98		0.17
POS	38.3		29.8	
NEG	56.6		63.8	
Unknown	5.1		6.4	
Exitus (%)	13.8	0.12	13.3	0.19
Relapse (%)	25.0	0.27	21.0	0.15

N: Sample size; SD: Standard deviation; TNBC: Triple negative subtype; ER: Estrogen receptor; PR: Progesterone receptor; BCVY: Breast cancer in very young women; BCO: Breast cancer in older women. P-values indicate statistics for the differences among HDAC5 expression and the different clinicopathological (CP) features included in the table by age groups. * *p*-value < 0.05, ** *p*-value < 0.01.

**Table 2 cancers-12-00412-t002:** Cell lines characteristics and culture conditions.

Cell Line	Subtype	Receptor Expression	Tumor Type	Age	Culture Medium	Conditions	Supplements
HCC 1500	Luminal	ER, PR	IDC	32	RPMI	5% CO_2_ 37 °C	1% L-glu 10% FBS
HCC1937	Basal	EGP2	IDC	24	RPMI	5% CO_2_ 37 °C	1% L-glu 10% FBS
MDA-MB-231	Basal	EGFR, TGF-β	Carcinoma	51	RPMI	5% CO_2_ 37 °C	1% L-glu 10% FBS
MCF7	Luminal	ER, IGFBP	IDC	69	RPMI	5% CO_2_ 37 °C	1% L-glu 10% FBS
BT474	Luminal	ER, PR, HER2	IDC	60	DMEM	5% CO_2_ 37 °C	1% L-glu 10% FBS
HCC1806	Basal	EGP2	Carcinoma	60	RPMI	5% CO_2_ 37 °C	1% L-glu 10% FBS 10% FBS

EGP2: Epithelial glycoprotein 2; EGFR: Epidermal growth factor receptor; TGF-β/α: transforming growth factor β/α; ER: Estrogen receptor; PR: Progesterone receptor; HER2: Hormonal estrogen receptor 2; IGFBP: Insulin growth factor binding protein; RPMI: RPMI 1640 medium; FBS: Fetal bovine serum; L-glu: L-glutamine; IDC: Invasive ductal carcinoma.

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
