# Peer review of "HDAC5 Inhibitors as a Potential Treatment in Breast Cancer Affecting Very Young Women"

_cancers, 2020, doi:10.3390/cancers12020412_

Round 1

Reviewer 1 Report

The authors investigated HDAC5 expression in relation with clinical and pathological characteristics of two cohorts of breast cancer: one from women <35 (60 cases, BCVY) and the other from women >45 (47 cases, BCO). They also studied the effects of the HDAC5 inhibitor LMK-235 on breast cancer cell lines derived both from young and older individuals. It is a pity that more BCO cases, easier to obtain, have not been tested; should have at least analysed the same number of cases tested for BCVY to have a 1:1 comparison. In addition, since a high percentage of breast cancers arising in young women have a hereditary origin, they do not specify if familial cases (mainly BRCA1 or BRCA2 mutant) where excluded from the study (or included). Hereditary breast cancer has specific clinical, biological and molecular characteristics different from sporadic breast cancer, and also treatment response is different. Of note, the HCC1937 cell line they use to represent BCVY originates from a BRCA1-mutant patient and HCC1500 cells are from a patient with a significant family history of early-onset colon cancer and with a sister with breast cancer (see its characteristics at ATCC). They could not be representative models of the BCVY cases analysed in the study. Most likely, short term cultures, or even 3D cultures derived from the tissue specimens of BCVY, should used as models.

There are many inconsistencies between the figures/tables and the text. In some parts authors mention results that are not shown (lines 95 and 186); these data must be presented at least as supplementary material. Many results are overestimated and some data are missing, leading to suspect that only data that support the hypothesis formulated by the authors have been presented.

There is no correspondence between the percentages of the histological subtypes indicated in the text (lanes 79-82) and those shown in table 1. They must be corrected.

From the boxplots in figure 1A the distribution of the BCVY and BCO is very similar and the significance of the p-value is borderline (0.04). As for figure Figure 1B, are “no exitus” and “exitus” groups balanced? How many cases/class had “exitus”? Add this information. It is quite unusual that HDAC5 levels have an opposite trend respect to their status in the two groups. The result could have been biased by the unbalance of the cases analysed in the two groups.

Figure S1 shows that the subtypes expressing higher HDAC5 for both BCVY and BCO are luminal A/B (not HER2, see line 96 and luminal/HER2, see line 97). The sentence at lines 97-99 must be changed accordingly.

Regarding the association between HDAC5 and survival, the criteria used for dividing cases into three groups according to expression of HDAC5 are not indicated (line 106-107). Was the choice of the cut-off values based on previous knowledge? If the authors chose the cut-off based on the observed association between the expression of the HDAC5 and the outcome, the reported results would not be valid (overestimation of the real association). A better categorization should be obtained by determining cut-off values using tertiles. Alternatively, the association between the expression of HDAC5 and the clinical characteristics of cases must be done using HDAC5 levels as continuous variable. This type of analysis would guarantee a better statistical power compared to the analysis that uses the categorized values of HDAC5". Please explain the choice or repeat the analysis as previously suggested.

As for the experiments on cells lines, LMK-235 significantly reduces viability on basal-like breast cancer cell lines and partially on HCC1500 cells after 72 hours of treatment. However, a single model of “young” breast cancer has been investigated. The results must be replicated on other models of “young” breast cancer, such as for example short term cultures or even 3D cultures derived from tissue specimens from BCVY. Finally, a colony formation assay would better investigate the effects of LMK-235 on these cells.

Regarding the wound healing assay, a picture showing the complete heal of the wound (and the time when it occurred) must be added for all the cell lines tested. Authors should also discuss the migrating cells that are visible in the wound of the three LMK-235 treated luminal cell lines. Finally, I cannot see the 5% reduction they mention for HCC1500 cells. These cells show high levels of early apoptosis also when treated with DMSO (fig. 5A). I wonder if the growth conditions of these cells are correct.

The plot in figure 4B does not reflect data shown in figure 4A (for example untreated HCC1806 cells do not have 100% wound closure at 48h, while HCC1937 have).

To be consistent with data, add HDAC5 mRNA expression levels for BT474 cells in Fig 5B, as well as HCC1500 and HCC1806 accumulation of acetyl-H3 in Fug 5C, and discuss the results according to the levels of HDAC5 mRNA and accumulation of acetyl-H3 observed in the missing data. As for western blotting, the densitometric analysis shows that accumulation of acetyl-H3 occurs only in MDA-MB-231 and MCF-7 cells, but this result is different from what stated in lines 189-193.

Author Response

Dear Sir/ Madam,

Many thanks for your comments. All of them are really interesting. We are going to answer point by point:

The authors investigated HDAC5 expression in relation with clinical and pathological characteristics of two cohorts of breast cancer: one from women <35 (60 cases, BCVY) and the other from women >45 (47 cases, BCO). They also studied the effects of the HDAC5 inhibitor LMK-235 on breast cancer cell lines derived both from young and older individuals. It is a pity that more BCO cases, easier to obtain, have not been tested; should have at least analysed the same number of cases tested for BCVY to have a 1:1 comparison.

We agree with your comment. Initially the number of samples were 1:1, however, part of BCO samples presented bad quality and were remove for the study. We consider that 47 is a enough size to get statistical results. Additionally, the number of patients is not extremally different and is one of the firsts studies with high sample size of BC patients younger than 35 years old.

In addition, since a high percentage of breast cancers arising in young women have a hereditary origin, they do not specify if familial cases (mainly BRCA1 or BRCA2 mutant) where excluded from the study (or included). Hereditary breast cancer has specific clinical, biological and molecular characteristics different from sporadic breast cancer, and also treatment response is different. Of note, the HCC1937 cell line they use to represent BCVY originates from a BRCA1-mutant patient and HCC1500 cells are from a patient with a significant family history of early-onset colon cancer and with a sister with breast cancer (see its characteristics at ATCC). They could not be representative models of the BCVY cases analyzed in the study. Most likely, short term cultures, or even 3D cultures derived from the tissue specimens of BCVY, should used as models.

We apologize for the missing information about hereditary origin and we have added it in the main text (line 79). We excluded in the study BC samples with BRCA1 or/and BRCA2 mutations in order to avoid confounding factors. Regarding the breast cancer cell lines, there is a limitation in the number of cell lines from young patients commercially available, is for that reason that we are considering to work in 3D organoids for future studies.  However, HCC1937 and HCC1500 cell lines have been used in previous group studies and have been compared with BC very young samples and both have presented a similar expression profile (Peña-Chilet et al BMC cancer 2014).  

There are many inconsistencies between the figures/tables and the text. In some parts authors mention results that are not shown (lines 95 and 186); these data must be presented at least as supplementary material. Many results are overestimated and some data are missing, leading to suspect that only data that support the hypothesis formulated by the authors have been presented.

We initially considered that these results were not relevant for the main study, however, now, we included them as a supplemental material in Figure S1A.

There is no correspondence between the percentages of the histological subtypes indicated in the text (lanes 79-82) and those shown in table 1. They must be corrected.

Thank you very much for the annotation we apologize for the mistakes and we have corrected them in the main text.

From the boxplots in figure 1A the distribution of the BCVY and BCO is very similar and the significance of the p-value is borderline (0.04). As for figure Figure 1B, are “no exitus” and “exitus” groups balanced? How many cases/class had “exitus”? Add this information. It is quite unusual that HDAC5 levels have an opposite trend respect to their status in the two groups. The result could have been biased by the unbalance of the cases analysed in the two groups.

The groups are balance being a 13.8% (n= 9) of exitus samples in BCVY and 13.3% (n= 6) of exitus in BCO, and the same for no exitus: BCVY (n=51), BCO (n=41). This information is included in the Table 1. However, we agree with reviewer 1 and we have added it in the main text (lines 102-103)

Figure S1 shows that the subtypes expressing higher HDAC5 for both BCVY and BCO are luminal A/B (not HER2, see line 96 and luminal/HER2, see line 97). The sentence at lines 97-99 must be changed accordingly.

We appreciate your comment and maybe is not accurately explained in the text. In the main text we refer to the HDAC5 expression between BCVY and BCO for the same subtypes. Thus, we observed higher HDAC5 expression in Luminal B and HER2 tumours from BCVY in comparison with BCO patients from the same subgroups. While BCO patients presented higher expression for Luminal A and Luminal/HER2 comparing to BCVY. We have changed the sentence in the manuscript.

Regarding the association between HDAC5 and survival, the criteria used for dividing cases into three groups according to expression of HDAC5 are not indicated (line 106-107). Was the choice of the cut-off values based on previous knowledge? If the authors chose the cut-off based on the observed association between the expression of the HDAC5 and the outcome, the reported results would not be valid (overestimation of the real association). A better categorization should be obtained by determining cut-off values using tertiles. Alternatively, the association between the expression of HDAC5 and the clinical characteristics of cases must be done using HDAC5 levels as continuous variable. This type of analysis would guarantee a better statistical power compared to the analysis that uses the categorized values of HDAC5". Please explain the choice or repeat the analysis as previously suggested.

We appreciate this comment, the criteria used for the classification of the samples according to the HDAC5 expression is included in Material & Methods section (lines 331-332). Representing HDAC5 expression as a continuous variable we have stablish three cut-off in order to obtain three expression group levels.

As for the experiments on cells lines, LMK-235 significantly reduces viability on basal-like breast cancer cell lines and partially on HCC1500 cells after 72 hours of treatment. However, a single model of “young” breast cancer has been investigated. The results must be replicated on other models of “young” breast cancer, such as for example short term cultures or even 3D cultures derived from tissue specimens from BCVY. Finally, a colony formation assay would better investigate the effects of LMK-235 on these cells.

We appreciate your comments. As we mention before, we are considering the used of 3D organoids in order to obtain a better BCVY model and it would be published in future studies.

Regarding the wound healing assay, a picture showing the complete heal of the wound (and the time when it occurred) must be added for all the cell lines tested. Authors should also discuss the migrating cells that are visible in the wound of the three LMK-235 treated luminal cell lines. Finally, I cannot see the 5% reduction they mention for HCC1500 cells. These cells show high levels of early apoptosis also when treated with DMSO (fig. 5A). I wonder if the growth conditions of these cells are correct.

We fully agree with your comment. So, we have modified some of the WH assay images. All cell lines are treated under the same conditions. HCC1500 cell line showed lower grow rates in comparison with the other cell lines even under normal/no dmso conditions. The DMSO WH migration rates are similar to the no dmso migration rates. So, we could say that the growth conditions were correct.

To be consistent with data, add HDAC5 mRNA expression levels for BT474 cells in Fig 5B, as well as HCC1500 and HCC1806 accumulation of acetyl-H3 in Fug 5C, and discuss the results according to the levels of HDAC5 mRNA and accumulation of acetyl-H3 observed in the missing data. As for western blotting, the densitometric analysis shows that accumulation of acetyl-H3 occurs only in MDA-MB-231 and MCF-7 cells, but this result is different from what stated in lines 189-193.

As the reviewer 1 suggest, we have considered to repeated the western blot studies (Figure 5C) and we have corrected and improve the results in the main text and additionally we have included the densitometry study in the Figure 5D which shows that the accumulation of acetyl-H3 occurs in all breast cancer cell lines analysed, supporting the correct function of the LMK-235 inhibitor.

Reviewer 2 Report

Oltra et al. have evaluated the association between HDAC5 overexpression and intrinsic subtype in women with varying ages of onset. Having found a negative correlation between HDAC5 expression and survival in the early onset group they then evaluated the use of LMK-235, a specific HDAC5 inhibitor in a panel of breast cancer cell lines. The inhibitor showed differential efficacy on the various cell lines in regards to both cytotoxicity and migration.

The negative correlation between HDAC5 overexpression and overall survival has been previously described by numerous others. But, the authors by studying the differences in HDAC5 expression and its prognostic value between tumors of similar histotypes, and stratifying for age at onset they have added a new level of detail which is of value to the community.

While the stratification is novel many of the other observations could be further strengthened by addressing the following items:

Figure 1B the individual data points need to be added to the figure. Otherwise, as Figure 1B and Figures 2A and B are actually different representations of the same data: the correlation between HDAC5 expression and survival in BCO versus BCVY, one could move 1B to supplementary and merge figures 1 and 2. Figure 2: LMK235 has previously been evaluated in several breast cancer cell line panels, so the novelty here is extremely low. An evaluation of the efficacy of this drug in a more physiological setting, such as organoid cultures (seeding your cells in BME) could add an additional level of information. Figure 4: is it possible to say that a drug does not have an effect on a population of cells which are not migrating, and is the case for BT474? I would think not. Furthermore, although the quality of the images makes it difficult, it actually appears that there is increased scattering in BT474_LMK235, HCC1500_LMK235 and MCF7_LMK235 that does not correlate at all with the limits that have been delimitated by the authors. Could HDAC inhibition correlate with increased scattering in certain cases? Another point is that would healing is a collective migration phenotype. Would the same cells subjected to treatment in random migration be similarly affected? Figure 5: The control for the western in C. should not be GAPDH: a largely cytosolic protein of the glycolytic cascade which is meaningless in this setting, but the pan H3. Inclusion of a panel for HDAC5 protein expression would be informative in this setting. Have the authors considered the previously published demonstrating a functional tumor-promoting interaction between HDAC5 and LSD1, and do LSD expression patterns follow similar patterns in the BCO versus BCVY data-sets? This could help to provide you with a mechanism which is currently lacking.

Overall, the findings, if substantiated further will present a new perspective to the field.

Author Response

Dear Sir/ Madam,

Many thanks for your comments. All of them are really interesting. We are going to answer point by point:

Figure 1B the individual data points need to be added to the figure.

Thank you very much for the annotation, we have improved the Figure 1 following your advice.

Otherwise, as Figure 1B and Figures 2A and B are actually different representations of the same data: the correlation between HDAC5 expression and survival in BCO versus BCVY, one could move 1B to supplementary and merge figures 1 and 2.

Many thanks for your suggestions. Although we find them interesting, we think that maintain the figures in the same position could emphasize the relationship between HDAC5 expression and survival.  In our opinion it is important to demonstrate this relationship in both figures.

Figure 2: LMK235 has previously been evaluated in several breast cancer cell line panels, so the novelty here is extremely low. An evaluation of the efficacy of this drug in a more physiological setting, such as organoid cultures (seeding your cells in BME) could add an additional level of information.

We agree with your comment and we have considered the use of organoids 3D models for future studies.

Figure 4: is it possible to say that a drug does not have an effect on a population of cells which are not migrating, and is the case for BT474? I would think not. Furthermore, although the quality of the images makes it difficult, it actually appears that there is increased scattering in BT474_LMK235, HCC1500_LMK235 and MCF7_LMK235 that does not correlate at all with the limits that have been delimitated by the authors. Could HDAC inhibition correlate with increased scattering in certain cases? Another point is that would healing is a collective migration phenotype. Would the same cells subjected to treatment in random migration be similarly affected?

Many thanks for your comments. As you suggest, the HDAC inhibitor has not the same effect in all cell line populations, affecting mainly TN subtypes and younger cell lines independently of their molecular subtype (HCC1500 and HCC1937). However, the effect is lower or unappreciable for luminal cell lines from older patients, such as BT474 and MCF7. Additionally, wound healing assays have been performed by triplicates, so we consider that they are robust. We hope we have answered your question.

Figure 5: The control for the western in C. should not be GAPDH: a largely cytosolic protein of the glycolytic cascade which is meaningless in this setting, but the pan H3. Inclusion of a panel for HDAC5 protein expression would be informative in this setting.

That is a good appreciation and we have repeated the western blot studies (Figure 5C) in order to improve the control and additionally we have included the densitometry study in the Figure 5D which shows that the accumulation of acetyl-H3 occurs in all breast cancer cell lines analysed, supporting the correct function of the LMK-235 inhibitor.

Have the authors considered the previously published demonstrating a functional tumor-promoting interaction between HDAC5 and LSD1, and do LSD expression patterns follow similar patterns in the BCO versus BCVY data-sets? This could help to provide you with a mechanism which is currently lacking.

We appreciate your comment. That is a really good point and we will consider it in the following studies to go in depth in the mechanism.

Reviewer 3 Report

The paper by Oltra et al. hypothesized that HDAC5 inhibitor LMK-235 may have a different activity depending on breast cancer patients age. To test their hypothesis, they analyzed the expression of HDAC5 expression in breast cancer samples from very young women (BCVY) and older patients (BCO) and their correlation with clinical features. In addition, they studied the potential of HDAC5 inhibition in BC cell lines of different molecular subtypes and obtained from women of different age, very young or old.

The study design is innovative and shows that LMK-235 acts differently depending on age of patients and BC phenotypes. The limit, as stated by the authors is on the relatively small number of patients analyzed that unfortunately limited the statistical results that do not reach significance.

Results presentation could be rendered more understandable by adding old or young near the cell lines (fig. 3).

Author Response

Dear Sir/ Madam,

Many thanks for your comments. As the reviewer indicates, the number of samples is limited. However, BC affecting very young samples represents around 5% of all breast cancers. For that reason, our study represents one of the first with a considerable number of BC from very young patients.

We fully agree with the suggestion of the reviewer 3, so following your recommendations, we added the age in the Figure 3.

Reviewer 4 Report

In this manuscript, Oltra et al., demonstrated that high expression of HDAC5 is associated with poor survival outcome in very young breast cancer patients (BCVY). Such an association was not found in old breast cancer patients (BCO). They then proposed that inhibition of HDAC5 should be able to suppress certain features of cancer cells i.e. cell proliferation and migration. Through the use of LMK-235, they found that the responses were different in various breast cancer cells. Subsequently, the authors demonstrated that treatment with LMK-235 could increase the proportion of apoptotic cells, enhance the level of acetyl-histone H3 and increase mRNA expression of HDAC5. The findings however do not really address how the high HDAC5 expression would contribute to the molecular mechanism of breast cancer pathogenesis. In addition, some of the following concerns should be also be addressed.

Concerns:

For their in vivo study, they employed mRNA levels of HDAC5 to classify the patients and showed that high mRNA level of HDAC5 associated with poor survival outcome. This was followed by in vitro study on cell line in which they observed increased HDAC5 mRNA expression in all breast cancer cell lines treated with HDAC5 inhibitor LMK-235 which they maintained would restore protein expression. However they found HDAC5 protein was too low to be measured. Hence they resorted to measuring HDAC5 activity by determining the levels of acetyl-histone H3. This they showed was increased both for luminal as well as triple negative breast cancer. Hence it is not entirely clear the significance of mRNA levels of HDAC5. The in vitro results and in vivo results don’t really have anything to do with each other. It unclear whether the breast cancer samples collected for analysis were primary breast cancer samples from patients who had not received any prior treatment, as this might alter the molecular profile of the samples. LMK-235 is known to be able to target different HDACs such as HDAC1, 2, 4, 6 etc at nM level. In their in vitro study, the authors employed 20 uM of LMK-235. Therefore, it was very hard to conclude whether inhibition of HDAC5 plays any major role in suppressing cell viability. The off-target effect would also be a problem, especially since it was mentioned that LMK-235 was the most cytotoxic compound amongst the new HDAC inhibitors identified. In their in vitro study, the authors did not verify the mRNA and protein expression of HDAC5 in their cell lines used. In their cell viability study, ER+ve cell lines MCF7 (BCO), BT474 (BCO), HCC1500 (BCVY) did not respond to LMK-235. Would it be possible that the presence of ER could compromise the effect of LMK-235? In figure 5a, the author treated the cells with different concentrations of LMK-235 for 48 hours and performed apoptosis assay. They found that the treatment of LMK-235 could induce apoptosis. These results do not tally with the findings shown in figure 3a as whilst significant increase in apoptotic cells was seen for MCF7 and HCC1500, figure 3a did not show expected reduction of cell viability. The results were not consistent. The authors should check their data.

Minor:

The quality of the western blot result is bad especially the loading control GAPDH. It would be good to indicate the number of cases in each of the groups (Figure 1 and Figure 2.)

Author Response

Dear Sir/ Madam,

Many thanks for your comments. All of them are really interesting. We are going to answer point by point:

For their in vivo study, they employed mRNA levels of HDAC5 to classify the patients and showed that high mRNA level of HDAC5 associated with poor survival outcome. This was followed by in vitro study on cell line in which they observed increased HDAC5 mRNA expression in all breast cancer cell lines treated with HDAC5 inhibitor LMK-235 which they maintained would restore protein expression. However they found HDAC5 protein was too low to be measured. Hence they resorted to measuring HDAC5 activity by determining the levels of acetyl-histone H3. This they showed was increased both for luminal as well as triple negative breast cancer. Hence it is not entirely clear the significance of mRNA levels of HDAC5. The in vitro results and in vivo results don’t really have anything to do with each other

We think that the reviewer 4 is pointing to the relation between HDAC5 expression at mRNA level and protein level. As we mention in the study, the HDAC5 inhibitor is reducing the HDAC5 protein activity (as WB indicates). Nevertheless, this reduction at the protein level is inducing a positive feedback increasing the mRNA HDAC5 expression, which is observed at mRNA expression results. Although their function is at protein level, the used of this inhibitor could reduce the HDAC5 protein activity in the in-vivo and could be a potential treatment in BCVY patients. Additionally, although the protein levels are increased in all subtypes, HDAC5 has not the same effect in all breast cancer subtypes as the patients results showed.

The breast cancer samples collected for analysis were primary breast cancer samples from patients who had not received any prior treatment, as this might alter the molecular profile of the samples

All samples were collected before any treatment, with the intention of minimizing any mistake in the results.

The off-target effect would also be a problem, especially since it was mentioned that LMK-235 was the most cytotoxic compound amongst the new HDAC inhibitors identified. In their in vitro study, the authors did not verify the mRNA and protein expression of HDAC5 in their cell lines used.

This is a good comment and we though as you, so for that reason the mRNA and protein expression analysis in the cell lines under DMSO control conditions are indicated in the Figure 5b and 5c.

ER+ve cell lines MCF7 (BCO), BT474 (BCO), HCC1500 (BCVY) did not respond to LMK-235. Would it be possible that the presence of ER could compromise the effect of LMK-235?

That is a really good appreciation and a possibility that we have discussed in the main text “Interestingly, response was lower in ER positive cell lines, with the exception of HCC1500 which presented an intermedium response in comparison with the rest of ER positive cells. Previous studies found LMK-235 response in triple negative BC cell lines [33-36]. However, our results showed an important response to LMK-235 in breast cancer cell lines from young women that present luminal subtype, as HCC1500. These results suggest an effect of ER in the LMK-235 treatment but must exist other mechanisms in young cell lines that increase the effect of LMK-235 which are not present in older cell lines” (lines 249-255). ER+ could be implicated in the LMK treatment response. Nevertheless, we found really interesting the major effect of the treatment in the luminal young cell lines (HCC1500) in comparison with ER+ older cell lines (MCF7 and BT474).

These results do not tally with the findings shown in figure 3a as whilst significant increase in apoptotic cells was seen for MCF7 and HCC1500, figure 3a did not show expected reduction of cell viability. The results were not consistent. The authors should check their data.

Many thanks for your comment. HCC1500 cell line presents a slower grow rates, and the reduction in the viability assay is lower in comparison with the other cell lines, but we can observe a reduction around 50%. In the other hand, the increase in the apoptosis after LMK235 treatment could be also observed at the WH assay where the increase in the apoptosis is reflected in the increase in the space between cells due to the apoptosis experienced. Nevertheless, results were checked, and we found a mistake in MCF7 for the DMSO condition, where we represented wrong data. Apoptosis values were similar between DMSO and LMK-235 treatment at different conditions with the exception of high doses of LMK-235 were apoptosis was around 12%, however these differences were not statistically significant. We apologize for the mistake and now data is corrected.

The quality of the western blot result is bad especially the loading control GAPDH.

We agree with the comment and we have repeated the western blot (Figure 5C) and additionally we have included the densitometry study in the Figure 5D which shows that the accumulation of acetyl-H3 occurs in all breast cancer cell lines analysed, supporting the correct function of the LMK-235 inhibitor.

It would be good to indicate the number of cases in each of the groups (Figure 1 and Figure 2.) 

We thank the reviewer 2 for this comment, we are in complete agreement with you in including this information in the main text (Figure 1 and Figure 2).

Round 2

Reviewer 1 Report

The authors investigated HDAC5 expression in relation with clinical and pathological characteristics of two cohorts of breast cancer: one from women <35 (60 cases, BCVY) and the other from women >45 (47 cases, BCO). They also studied the effects of the HDAC5 inhibitor LMK-235 on breast cancer cell lines derived both from young and older individuals. It is a pity that more BCO cases, easier to obtain, have not been tested; should have at least analysed the same number of cases tested for BCVY to have a 1:1 comparison.

We agree with your comment. Initially the number of samples were 1:1, however, part of BCO samples presented bad quality and were remove for the study. We consider that 47 is a enough size to get statistical results. Additionally, the number of patients is not extremally different and is one of the firsts studies with high sample size of BC patients younger than 35 years old.

R: I do not agree with this comment, a 1:1 design would have given a stronger value to the study. Sporadic BC is very common and not so difficult to collect. Authors should have added new cases to the study. This is a limit for their analysis

In addition, since a high percentage of breast cancers arising in young women have a hereditary origin, they do not specify if familial cases (mainly BRCA1 or BRCA2 mutant) where excluded from the study (or included). Hereditary breast cancer has specific clinical, biological and molecular characteristics different from sporadic breast cancer, and also treatment response is different. Of note, the HCC1937 cell line they use to represent BCVY originates from a BRCA1-mutant patient and HCC1500 cells are from a patient with a significant family history of early-onset colon cancer and with a sister with breast cancer (see its characteristics at ATCC). They could not be representative models of the BCVY cases analyzed in the study. Most likely, short term cultures, or even 3D cultures derived from the tissue specimens of BCVY, should used as models.

We apologize for the missing information about hereditary origin and we have added it in the main text (line 79). We excluded in the study BC samples with BRCA1 or/and BRCA2 mutations in order to avoid confounding factors. Regarding the breast cancer cell lines, there is a limitation in the number of cell lines from young patients commercially available, is for that reason that we are considering to work in 3D organoids for future studies. However, HCC1937 and HCC1500 cell lines have been used in previous group studies and have been compared with BC very young samples and both have presented a similar expression profile (Peña-Chilet et al BMC cancer 2014).

R: I do not agree with this comment, as also authors state, BRCA1 or/and BRCA2 mutations could be confounding factors in their studies.I cannot see the mentioned comparison in Peña-Chilet et al BMC cancer 2014

We appreciate this comment, the criteria used for the classification of the samples according to the HDAC5 expression is included in Material & Methods section (lines 331-332). Representing HDAC5 expression as a continuous variable we have stablish three cut-off in order to obtain three expression group levels. 

R: The text at lines 331-332 is not clear: the criteria used for the classification of the samples according to the HDAC5 expression MUST BE INDICATED IN THE TEXT

As for the experiments on cells lines, LMK-235 significantly reduces viability on basal-like breast cancer cell lines and partially on HCC1500 cells after 72 hours of treatment. However, a single model of “young” breast cancer has been investigated. The results must be replicated on other models of “young” breast cancer, such as for example short term cultures or even 3D cultures derived from tissue specimens from BCVY. Finally, a colony formation assay would better investigate the effects of LMK-235 on these cells.

We appreciate your comments. As we mention before, we are considering the used of 3D organoids in order to obtain a better BCVY model and it would be published in future studies.

R: The results MUST be replicated on other models of “young” breast, please do this. 

Regarding the wound healing assay, a picture showing the complete heal of the wound (and the time when it occurred) must be added for all the cell lines tested. Authors should also discuss the migrating cells that are visible in the wound of the three LMK-235 treated luminal cell lines. Finally, I cannot see the 5% reduction they mention for HCC1500 cells. These cells show high levels of early apoptosis also when treated with DMSO (fig. 5A). I wonder if the growth conditions of these cells are correct.

We fully agree with your comment. So, we have modified some of the WH assay images. All cell lines are treated under the same conditions. HCC1500 cell line showed lower grow rates in comparison with the other cell lines even under normal/no dmso conditions. The DMSO WH migration rates are similar to the no dmso migration rates. So, we could say that the growth conditions were correct.

R: the assay was not modified as requested, no discussion on the results LMK-235 cells was done

Author Response

The authors investigated HDAC5 expression in relation with clinical and pathological characteristics of two cohorts of breast cancer: one from women <35 (60 cases, BCVY) and the other from women >45 (47 cases, BCO). They also studied the effects of the HDAC5 inhibitor LMK-235 on breast cancer cell lines derived both from young and older individuals. It is a pity that more BCO cases, easier to obtain, have not been tested; should have at least analysed the same number of cases tested for BCVY to have a 1:1 comparison.

Author previous reply:

We agree with your comment. Initially the number of samples were 1:1, however, part of BCO samples presented bad quality and were remove for the study. We consider that 47 is a enough size to get statistical results. Additionally, the number of patients is not extremally different and is one of the firsts studies with high sample size of BC patients younger than 35 years old.

R: I do not agree with this comment, a 1:1 design would have given a stronger value to the study. Sporadic BC is very common and not so difficult to collect. Authors should have added new cases to the study. This is a limit for their analysis

Author Reply:

We first calculated sample size for a study between two independent averages. Level of confidence (alpha risk) was set α = 0.05, assuming a level of 95% confidence, in a bilateral contrast (two-sided). Beta risk (the power of the analysis), was set at 0.2 (power of 80%). When defining the ratio between two groups, we assumed same initial number of patients from the two groups (young vs old; ratio 1). As we were performing the calculation based on means, we set the standard deviation (STD) in 3, taking into account that patient samples carry on high heterogeneity. Finally, we considered the minimum difference between groups to be detected as 2 folds (arbitrary units), and 20% (0.2) expected proportion of loosed samples (bad quality of RNA). According to these criteria, we obtained a number of 45 patients in each group (young vs old) to perform our analysis of HDAC5 mRNA expression. Link to our Sample Size Calculation system is https://www.imim.es/ofertadeserveis/software-public/granmo/. However, due both to capability of our team, as well as the availability of the samples, we were able to increase the total size of the two groups, reaching 60 samples from young patients and 47 samples from the old ones.

Related to the statistical analysis of the samples, the Wilcoxon rank sum test can be used with unequal sample size. Henry et al 1947, in Annals of Mathematical Statistic showed the consistency of the results for different sample size and also for small samples "On a Test of Whether one of Two Random Variables is Stochastically Larger than the Other".

Additionally, although sporadic breast cancer is very common and not so difficult to collect the, extraction, evaluation and reanalysis of the results is a long process and we found it difficult to address in 10 days.  

In addition, since a high percentage of breast cancers arising in young women have a hereditary origin, they do not specify if familial cases (mainly BRCA1 or BRCA2 mutant) where excluded from the study (or included). Hereditary breast cancer has specific clinical, biological and molecular characteristics different from sporadic breast cancer, and also treatment response is different. Of note, the HCC1937 cell line they use to represent BCVY originates from a BRCA1-mutant patient and HCC1500 cells are from a patient with a significant family history of early-onset colon cancer and with a sister with breast cancer (see its characteristics at ATCC). They could not be representative models of the BCVY cases analyzed in the study. Most likely, short term cultures, or even 3D cultures derived from the tissue specimens of BCVY, should used as models.

Author previous reply:

We apologize for the missing information about hereditary origin and we have added it in the main text (line 79). We excluded in the study BC samples with BRCA1 or/and BRCA2 mutations in order to avoid confounding factors. Regarding the breast cancer cell lines, there is a limitation in the number of cell lines from young patients commercially available, is for that reason that we are considering to work in 3D organoids for future studies. However, HCC1937 and HCC1500 cell lines have been used in previous group studies and have been compared with BC very young samples and both have presented a similar expression profile (Peña-Chilet et al BMC cancer 2014).

R: I do not agree with this comment, as also authors state, BRCA1 or/and BRCA2 mutations could be confounding factors in their studies.I cannot see the mentioned comparison in Peña-Chilet et al BMC cancer 2014

Author Reply:

We apologize for the confusion; the mentioned comparison is in an article that is under review in Journal of Cellular and Molecular Medicine. We can not include the results because of conflict of interest but, we can affirm, in view of our results, that the comparison of these young cell lines and tissue patients in terms of gene expression showed similar results between cell lines and patient samples from the same age groups.

It is completely true that both HCC1937 and HCC1500 are not exactly suitable models of Young Breast Cancer, since the first one contains BRCA1 mutation, and the second one has family history of early-onset colon cancer as well as a sister with breast cancer. However, as the research group is developing any other more suitable models, like primary cultures from patients, as well as 3D organoids, we thought that these two cell lines were the closer “initial” in vitro model to test our hypothesis (assuming BRCA1 mutation).  Anyway, we can affirm that the use of HCC1937 cell line (BCVY originates from a BRCA1-mutant patient) and HCC1500 cell line (unknown mutation in the BRCA genes) does not alter the results obtained.

Author previous reply:

We appreciate this comment, the criteria used for the classification of the samples according to the HDAC5 expression is included in Material & Methods section (lines 331-332). Representing HDAC5 expression as a continuous variable we have stablish three cut-off in order to obtain three expression group levels. 

R: The text at lines 331-332 is not clear: the criteria used for the classification of the samples according to the HDAC5 expression MUST BE INDICATED IN THE TEXT

Author Reply: 

Thank you very much for your appreciation, we apologize for the misunderstanding, maybe the numbers of lines have moved in the revision process and the lines are 352-354, additionally we have highlighted the text in yellow.

As for the experiments on cells lines, LMK-235 significantly reduces viability on basal-like breast cancer cell lines and partially on HCC1500 cells after 72 hours of treatment. However, a single model of “young” breast cancer has been investigated. The results must be replicated on other models of “young” breast cancer, such as for example short term cultures or even 3D cultures derived from tissue specimens from BCVY. Finally, a colony formation assay would better investigate the effects of LMK-235 on these cells.

Author previous reply:

We appreciate your comments. As we mention before, we are considering the used of 3D organoids in order to obtain a better BCVY model and it would be published in future studies.

R: The results MUST be replicated on other models of “young” breast, please do this. 

Author Reply:

Regarding of colony formation assay (long term proliferation) was initially considered, but as we obtained positive results in short term proliferation (MTT;24-72hrs), it was finally discarded.

Finally, thank you very much for your suggestion about the organoid models that have been fully taken into consideration but as we mentioned in the first revision, we are stablishing organoid 3D from young patients. However, the process is long, complicated and tricky and we found it impossible to address in 10 days. 

Regarding the wound healing assay, a picture showing the complete heal of the wound (and the time when it occurred) must be added for all the cell lines tested. Authors should also discuss the migrating cells that are visible in the wound of the three LMK-235 treated luminal cell lines. Finally, I cannot see the 5% reduction they mention for HCC1500 cells. These cells show high levels of early apoptosis also when treated with DMSO (fig. 5A). I wonder if the growth conditions of these cells are correct.

Author previous reply:

We fully agree with your comment. So, we have modified some of the WH assay images. All cell lines are treated under the same conditions. HCC1500 cell line showed lower grow rates in comparison with the other cell lines even under normal/no dmso conditions. The DMSO WH migration rates are similar to the no dmso migration rates. So, we could say that the growth conditions were correct.

R: the assay was not modified as requested, no discussion on the results LMK-235 cells was done 

Author Reply:

Regarding the first point, the WH assay showed the migration after 48h of the treatment. After that, some cell lines presented a complete heal of the wound such as HCC1806 control/DMSO, MDA-MB-231 control/DMSO and HCC1937-control/DMSO contrary to the treatment conditions where the migration was lower due to the treatment. Taking these considerations together, wound healing stop-point images were taken at time enough to observe the effects of the treatment in the wound closure, without compromising cell viability. These results agree with the cell viability assay, where the same cell lines presented a higher reduction in the viability under treatment conditions. The reduction in the viability correlates with a reduction in the migration. Additionally, HCC1500 cell line presented intermediated results in both studies. The 5% of reduction in the HCC1500 migration could be observed at the Figure 4B, where the percentage of migration was around 1% for the HCC1500 treated and ~6% for the HCC1500 DMSO/control. We have enlarged the results in the main text:

(Line 251)“Regarding that, we observed important viability reduction under treatment conditions for the cell lines that presented significant differences in the migration assay between control and treatment conditions. These results were observed in triple negative cell lines (from young and old patients). Thus, the reduction in the cell viability correlates with a reduction in cell migration under LMK-235 treatment. Interestingly, the luminal cell lines from old patients (BT474 and MCF7) were not affected by LMK-235 inhibitor. However, the luminal young cell line HCC1500 presented intermedium results between triple negative and luminal old cell lines. These results point out a potential breast cancer treatment not only for triple negative breast cancer but also for young patients from different molecular subtypes

Reviewer 4 Report

Most of the concerns have been adequated addressed except the following:

The results displayed in Figure 3 uses different colored symbols to distinguish the different cell lines, as indicated in the legend box. However the symbols are so small, and often covered by the error bars, that it becomes difficult to make out. It was with great difficulty to distinguish BT474 from HCC1500 which are both grey circles, only the former circle is larger, and in fact the data was mistaken on first reading. The authors now point out that HCC1500 gave intermediate response, so this must correspond to the third line in the graph. It would be best to actually label each line to faciltate easier reading. LMK-235 is known to be able to target different HDACs such as HDAC1, 2, 4, 6 etc at nM level. The lowest dose that gave significant difference for apoptosis for HCC1500 was 2.5 uM. At these levels, it is difficult to conclude whether inhibition of HDAC5 really does play a major role in suppressing cell viability. The off-target effect would also be a problem, especially since it was mentioned that LMK-235 was the most cytotoxic compound amongst the new HDAC inhibitors identified. These points should at least be discussed.

Author Response

Most of the concerns have been adequated addressed except the following:

The results displayed in Figure 3 uses different colored symbols to distinguish the different cell lines, as indicated in the legend box. However the symbols are so small, and often covered by the error bars, that it becomes difficult to make out. It was with great difficulty to distinguish BT474 from HCC1500 which are both grey circles, only the former circle is larger, and in fact the data was mistaken on first reading. The authors now point out that HCC1500 gave intermediate response, so this must correspond to the third line in the graph. It would be best to actually label each line to faciltate easier reading. LMK-235 is known to be able to target different HDACs such as HDAC1, 2, 4, 6 etc at nM level.

The lowest dose that gave significant difference for apoptosis for HCC1500 was 2.5 uM. At these levels, it is difficult to conclude whether inhibition of HDAC5 really does play a major role in suppressing cell viability. The off-target effect would also be a problem, especially since it was mentioned that LMK-235 was the most cytotoxic compound amongst the new HDAC inhibitors identified. These points should at least be discussed.

 Author Reply:

Many thanks for considering our effort to address all the comments, we considered all of them interesting to improve the paper. Following your suggestion we have improved the graph in order to facilitate de reading. We have removed the symbols and change some colour lines to an easy reading. Although LMK-235 has been described to inhibit HDACs 4, 5, and 6 at nM concentrations, 20µM concentration was selected in order to obtain a regulatory effect, taking into account that, in line with figure 3, this concentration was the only one capable of generating an effect on the proliferation of luminal cell lines in 48 hours. At the same time, the proliferation of TNBC lines was reduced by around 50%, allowing for an analysis. At longer times (72 hours), the viability of the TNBCs was seriously compromised. Anyway, we found interesting your appreciation about the HCC1500 cell lines and we have enlarged the discussion in the main text about it. We have included the following comments in the text:

Line (271)“It is worth to mention that LMK-235 treatment induces higher apoptosis in HCC1500 cell line at lower doses in comparison with the rest of cell lines. These results suggest the existence of off-target effects that increase the apoptosis at this cell line, so further studies are required to analyse the underlying mechanisms of LMK-235 inhibitor.

Round 3

Reviewer 1 Report

My criticism on the paper is not related to the sample size able to give significance (45), but to the fact that the comparison is unbalanced and the smallest group is that of sporadic cases (easiest to collect). I totally agree that 10 days of time is not enough to improve the paper, but this was not my suggestion. On the same hand I am convinced that the present sample size is not enough for a publication in this journal.

As for the cellular models used, I remain of the idea that the two lines used do not represent the BCVY (as also stated by the authors). Moreover I do not find acceptable the statement that “the use of HCC1937 cell line (BCVY originates from a BRCA1-mutant patient) and HCC1500 cell line (unknown mutation in the BRCA genes) does not alter the results obtained”. They are a different model.

Author Response

Dear reviewer 1,

Many thanks for your comments, all of them are constructive and have improved the article.

Regarding the first concern, we agree with you about that the perfect sample size should be an equal number of samples. However, the limitations in the collection and the quality of the samples after preprocessing have limited that point. However, it is worth to mention that this work is the first one that includes a study with an important amount of breast cancer samples from very young women. We have been working during the last years in this subgroup of patients and most of the papers represent a study of the clinicopathological features of these patients, but less is known about the molecular biology that occurs on them and there are few articles that analyze potential treatment options for them. For that reason, we considered that we have done a big effort to recruit this number of BCVY patients and study a potential treatment for them. Also in the scientific literature we find related articles in which 2 populations with different N are compared, for example, in 2015 in PLoS One, Muñoz-Rodriguez et al  perform an analysis  of  miRNAs in two groups: a) an early group representing women diagnosed with breast cancer ≤ 5.2 years postpartum (n = 12), and b) a late group representing women diagnosed with breast cancer ≥ 5.3 years postpartum (n = 44). Recently, in Oncologist 2020 January 31, Kogawa and cols try to determine whether the high HER2 FISH ratio is a predictor of pCR and prognosis in patients with non-metastatic inflammatory breast cancer and HER2+ non-IBC treated with neoadjuvant chemotherapy with or without trastuzumab. For that, they included 555 patients with stage III, HER+ breast cancer, 181 patients with IBC, and 374 with non-IBC, and analyze the results in the three cohorts, that present remarkable different sample sizes. Although, as we say, it is ideal to compare two populations with the same N, here we include some examples in which the slightly unbalance sample size does not imply that the results are wrong.

About the last concern, and in line with the previous mentioned, we think that the used of cell lines is the first step in the studies of new treatments. We are presenting a new potential treatment for BCVY patient based on the expression studies in patients and functional studies using the limited number of cell lines from young women commercially available. With all of them, we have observed interesting results about the HDAC5 inhibitor treatment. As you mention, the next step should be the study of it in 3D culture and we can affirm that we are stablishing 3D culture from young patients with success. However, it takes a long time and we are actively working on them to proceed with the study of potential treatments in 3D culture models. So, in future studies we will be able to corroborate our hypothesis in 3D models. 

Finally, taking into account the limitations we consider that the article has improved properly with the reviewer’s comments, which have reinforced our hypothesis about the differences between young and old patients with breast cancer tumors and the potential used of HDAC5 inhibitors in the future.